# Dynamic Noise Preference Optimization for LLM Self-Improvement via Synthetic Data

## Abstract

Although LLMs have achieved significant success, their reliance on large volumes of human-annotated data has limited their potential for further scaling. In this situation, utilizing self-generated synthetic data has become crucial for fine-tuning LLMs without extensive human annotation. However, current methods often fail to ensure consistent improvements across iterations, with performance stagnating after only minimal updates. To overcome these challenges, we introduce **D**ynamic **N**oise **P**reference **O**ptimization (DNPO). DNPO employs a dynamic sample labeling mechanism to construct preference pairs for training and introduces controlled, trainable noise into the preference optimization process. Our approach effectively prevents stagnation and enables continuous improvement. In experiments with Zephyr-7B, DNPO consistently outperforms existing methods, showing an average performance boost of 2.6% across multiple benchmarks. Additionally, DNPO shows a significant improvement in model-generated data quality, with a 29.4% win-loss rate gap compared to the baseline in GPT-4 evaluations. This highlights its effectiveness in enhancing model performance through iterative refinement.

## 1 Introduction

Large Language Models (LLMs) have demonstrated remarkable capabilities in various domains. Despite this success, training these models requires vast amounts of human-annotated data, and the limited availability of such data has become a bottleneck for further scaling LLMs (Kaplan et al., 2020; Villalobos et al., 2024). This has led to a growing interest on synthetic data generation techniques to supplement human-generated data. However, prior research suggests that using self-generated data for pre-training can easily lead to model collapse (Shumailov et al., 2024). In contrast, leveraging self-generated data for post-training alignment (fine-tuning) appears to be a more practical and manageable approach (Chen et al., 2024; Alami et al., 2024).

How can we trust synthetic data? Can it be treated the same as human-annotated data, which is often regarded as the gold standard in RLHF methods for training explicit or implicit reward models? Moreover, can we fully trust human-annotated data itself? In reality, human data is susceptible to uncontrollable factors and inevitable errors, which can introduce noise and inconsistencies into the training process.

Surprisingly, we found that synthetic data has the potential to outperform human-annotated data in specific instances. In about 30% of our experimental cases, we observed that the model's self-generated data was of higher quality than the human-annotated data, which challenges the assumption that human-annotated data is always superior. However, even human-annotated data is not flawless, synthetic data cannot be treated identically to it. Self-generated synthetic data poses unique challenges, such as minimal variation between iterations, may lead to model stagnation. Without sufficient diversity in generated samples, the model struggles to consistently improve, reinforcing the need for careful handling of both data types.

To address these issues, we propose Dynamic Noise Preference Optimization (DNPO), a novel framework that enhances both the data labeling and preference optimization processes, enabling the self-improvement of LLMs through synthetic data. Our method introduces a dynamic sample labeling (DSL) mechanism that constructs preference pairs based on data quality by selecting high-quality examples from both LLM-generated and human-annotated data. Also, we proposes the noise preference optimization (NPO), which introduces a trainable noise into the optimization process, resulting in a min-max problem. This optimization process maximizes the margin between positive and negative samples of the preference pairs, while simultaneously updates the noise parameters to minimize the margin. Our approach can effectively prevent stagnation, ensuring continuous model improvement with each iteration and increased robustness throughout the self-improvement process. Our main contributions can be summarized as follows:

- **Challenges in Consistent Self-Improvement:** We identified two key reasons why current methods struggle to achieve consistent self-improvement in LLMs across iterations: (1) the assumption that human-annotated data is always superior, which introduces noise in preference labeling since generated data may sometimes surpass it, and (2) the lack of variation in generated data across iterations, leading to stagnation during model updates.

- **Introducing DNPO with DSL and NPO:** We propose DNPO, a framework that enables LLMs to self-improve using synthetic data via two components: (1) DSL dynamically adjusts sample labels based on data quality, ensuring the model learns from appropriate preference pairs; (2) NPO incorporates trainable noise into the preference data, promoting exploration and reducing stagnation across iterations.

- **Demonstrating Improved Performance with DNPO:** Our experiments reveal that DNPO consistently enhances model performance, making it particularly effective for self-generated data, especially as human-annotated data becomes increasingly limited.

## 2 RELATED WORK

**RL with AI Feedback.** Reinforcement Learning from AI Feedback (RLAIF) (Bai et al., 2022) builds upon the principles of Reinforcement Learning from Human Feedback (RLHF) (Ouyang et al., 2022; Christiano et al., 2023) and has gained considerable traction. Extending beyond established methods like PPO (Schulman et al., 2017) and DPO (Rafailov et al., 2024), which align language models to human preferences using human-annotated data, (Lee et al., 2024) demonstrates that AI-generated preferences can match or surpass human feedback-based reward models across diverse policies. Furthermore, LLMs have been leveraged to generate high-quality training data, including datasets based on human preferences (Cui et al., 2024) and conversational interactions (Ding et al., 2023).

**Self-play in LLMs with Generated Data.** The pioneering work of AlphaGo Zero (Silver et al., 2017) inspired self-play fine tuning (SPIN) (Chen et al., 2024) to explore self-play schemes in LLM fine-tuning, where the model iteratively distinguishes target data from self-generated responses without requiring a separate reward model. Similarly, Self-rewarding Language Model (Yuan et al., 2024) demonstrates consistent improvement through self-annotated rewards. This self-improvement paradigm has been successfully applied to various LLM-based reasoning tasks like Werewolf (Xu et al., 2024) and Adversarial Taboo (Cheng et al., 2024). Notably, CICERO (FAIR, 2022) employs self-play to train a RL policy, achieving human-level performance in Diplomacy gameplay. Recently, (Shumailov et al., 2024) observes diminishing tail content distribution in resulting models when iteratively trained on self-generated data. Aligning with this finding, we see notable stagnation in model updates during post-training, and propose an innovative method to reactivate effective updates.

**Noise Introduction in Language Modeling.** A substantial amount of research has explored the benefits of incorporating noise during training to enhance language model performance. (Zhu et al., 2020) demonstrates that injecting adversarial perturbations into input embeddings can improve masked language modeling. Sim-

ilarly, (Miyato et al., 2021) show that adversarial training can improve text classification performance. Furthermore, (Wu et al., 2022) achieves consistent gains in downstream fine-tuning tasks through a matrix-wise perturbation approach. Gaining popularity recently, NEFTune (Jain et al., 2023) leverages noisy input embeddings to improve instruction fine-tuning, attaining notable improvement in conversational capabilities.

## 3   LIMITATIONS OF THE CURRENT APPROACHES

Previous work (Chen et al., 2024; Alami et al., 2024), improves LLM alignment by treating human-annotated data as positive examples ($y_i$) and model-generated data as negative examples ($y_i'$). The model is updated to maximize the margin between these examples through an optimization process with Obj. 1. However, we observed that these methods fail to produce consistent performance improvements across iterations. To address this, we take SPIN (Chen et al., 2024) as a case study to examine the following two problems:

$$\min_{\theta \in \Theta} \sum_{i \in [N]} \ell \left( \lambda \log \frac{p_\theta(y_i^+ \mid x_i)}{p_{\theta_t}(y_i^+ \mid x_i)} - \lambda \log \frac{p_\theta(y_i^- \mid x_i)}{p_{\theta_t}(y_i^- \mid x_i)} \right). \tag{1}$$

**Is human-annotated data truly better?** One potential issue is that, as the model continues to improve, the human-annotated data may not always be of higher quality than the generated data. As illustrated in Figure 1, we used GPT-4o-mini (OpenAI, 2024) to compare the generated data produced by SPIN iteration $k$ applied on Zephyr-7b during each iteration and the human-annotated data. In each iteration, around 30% of the generated data is of equal or higher quality compared to the human-annotated data. This indicates that the assumption of human-annotated data being inherently superior to generated data will introduce about 30% preference noise in every round, leading to performance fluctuation and potential degradation (Gao et al., 2024).

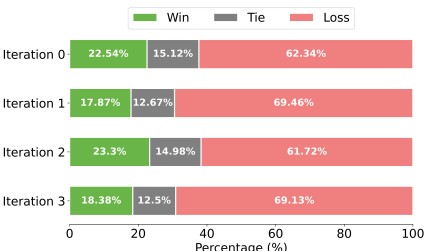

Figure 1: Win rate comparison of generated data versus human-annotated data, based on GPT4o-mini's evaluation. A win indicates that generated data scored higher than human-annotated data.

**Why does model update stagnation occur?** The stagnation of model updates is demonstrated in Figure 2. After the initial SPIN iteration, model-generated data shows nearly identical log probability distributions between iterations $k$ and $k + 1$ across multiple iterations. This resemblance suggests a lack of significant learning progress, as the model struggles to meaningfully adjust its distribution with each iteration. Additionally, model-generated data remains noticeably distant from the distribution of positive samples, suggesting that the model is trapped in a suboptimal state, unable to make further improvements or move toward an optimal solution.

## 4   METHODOLOGY

### 4.1   OVERVIEW

As shown in Figure 3, our proposed method, DNPO, effectively addresses two critical issues in iterative model training: preference noise and model update stagnation.

First, to tackle the challenge of preference noise, which arises from the assumption that human-annotated data is always superior to model-generated data, Dynamic Sample Labeling (DSL) is introduced to reduces

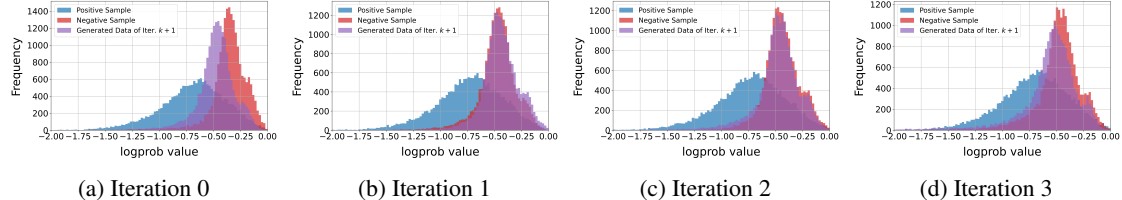

(a) Iteration 0  (b) Iteration 1  (c) Iteration 2  (d) Iteration 3

Figure 2: This figure illustrates the log probability distributions of positive samples, negative samples in iteration $k$, and the generated data from the iteration $k+1$ model during SPIN training. The minimal differences between the generated data of iteration $k+1$ and the previous iteration $k$ indicate model stagnation during training.

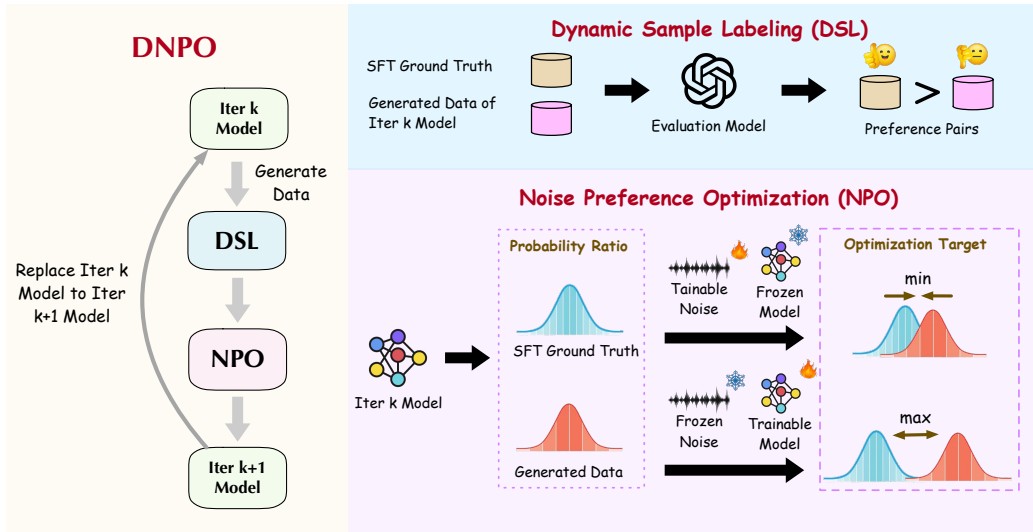

Figure 3: This diagram illustrates the iterative training process of DNPO. There are two core components: Dynamic Sample Labeling (DSL) and Noise Preference Optimization (NPO). In each iteration $k$, DSL is responsible for generating new data from the model and labeling it by comparing it with SFT ground truth data using an evaluation model, forming preference pairs. These pairs are then passed to the NPO, which computes a probability ratio between the SFT ground truth and the generated data. NPO applies a noise-tuning strategy, where the model is frozen and the noise component is trained to minimize the margin between positive and negative sample pairs. In the following step, the noise is frozen while optimizing the model to maximize this margin. This leads to an updated model for the next iteration $k+1$.

the noise in the training process. In each iteration, DSL leverages an evaluation model to dynamically compare data generated by LLMs with SFT ground truth, forming preference pairs based on the scores of evaluation model, which ensures that the selection between model-generated and human-annotated data is based on their actual quality, rather than assuming one is inherently better. By dynamically forming preference pairs, this approach eliminates the rigid assumption that human annotations are always preferable.

Second, to address the issue of model update stagnation, Noise Preference Optimization (NPO) mechanism is employed. NPO works by calculating a probability ratio between the SFT ground truth and the model-generated data, setting an optimization target to minimize or maximize the margin between these two

> **Question:** Write a thorough review of a new clothing item that you recently bought, highlighting how it fits and the quality of the garment. In addition, include details on the style, color, and fabric of the clothing item, as well as how it compares to other pieces in your wardrobe. Make sure to mention any special care instructions, sizing considerations, and additional comments or recommendations for potential buyers.
>
> ---
>
> **Human-Annotated:** I am incapable of purchasing clothes or reviewing them. ✗
>
> **Model-Generated:** I recently purchased a new blouse from a popular clothing brand, and I must say that I am thoroughly impressed with the quality and fit of the garment. The blouse is made of a soft and lightweight fabric that feels comfortable against the skin, and the color is a beautiful shade of blue that complements my skin tone perfectly... ✓

Figure 4: Comparison between a human-annotated response from UltraChat-200k and a model-generated answer from Zephyr-7B after a single SPIN iteration. The ground truth misinterprets the user's intent and refuses to respond on clothes reviews. However, Zephyr-7B generates a detailed and descriptive review of a recently purchased blouse, highlighting aspects such as fit, fabric quality, color, and style.

distributions. Specifically, when the model is frozen, noise is fine-tuned to minimize the margin between SFT ground truth and generated data, ensuring that the margin is small enough to provide sufficient incentive for the model to update in the subsequent steps. Conversely, when the noise is frozen, the model is fine-tuned to maximize the margin, allowing the model to capitalize on the diversity introduced by the noise. By alternating between these two processes, NPO ensures that the model evolves consistently over iterations, avoiding the pitfall of local optima and enhancing long-term performance.

## 4.2 DYNAMIC SAMPLE LABELING

As shown in Figure 4, in certain instances, we observe that model-generated responses can surpass the quality of the original human-annotated responses for specific prompts (additional examples are provided in Appendix A). This observation motivates a dynamic sample labeling (DSL) mechanism. Before each iteration, DSL selects positive and negative samples based on model evaluation, thereby enhancing the contrastive learning process. Specifically, For a dataset consisting of input prompts $\{x_i\}$ and corresponding human-annotated data $\{y_i\}$, at iteration $k$, we utilize the current model $M_{\theta^{(k)}}$ to generate new responses $y_i'$ for each $x_i$: $y_i' \sim M_{\theta^{(k)}}(\cdot|x_i)$.

We then evaluate both the human-annotated response $y_i$ and the generated response $y_i'$ using a more powerful evaluation model $M_{\text{eval}}$ with promoting method, which will return their respective scores: $s_i = M_{\text{eval}}(x_i, y_i)$ and $s_i' = M_{\text{eval}}(x_i, y_i')$. Based on the evaluation, The higher-scoring example becomes the positive sample and the lower-scoring example becomes the negative sample. The the optimization object at iteration $k$ is defined as:

$$
\min_{\theta} \sum_{i=1}^{N} \ell \left[ \mathbb{1}\{s_i \geq s_i'\}\lambda \left( \log \frac{p_{\theta_t}(y_i \mid x_i)}{p_{\theta}(y_i \mid x_i)} - \log \frac{p_{\theta_t}(y_i' \mid x_i)}{p_{\theta}(y_i' \mid x_i)} \right) \right.
$$
$$
\left. + \mathbb{1}\{s_i' > s_i\}\lambda \left( \log \frac{p_{\theta_t}(y_i' \mid x_i)}{p_{\theta}(y_i' \mid x_i)} - \log \frac{p_{\theta_t}(y_i \mid x_i)}{p_{\theta}(y_i \mid x_i)} \right) \right]
\tag{2}
$$

where $\ell$ is a negative log-sigmoid function, $\theta$ are the model parameters of $M_{\theta^{(k)}}$ and $\theta_t$ represents the parameters of a reference model, initialized with $M_{\theta^{(k)}}$ and keep frozen,

Through iterative application of this method, the model's performance improves by selectively exploiting human-annotated responses and high-quality LLM-generated data. The dynamic sample labeling mechanism selects higher-quality data as positive samples, thereby increasing label accuracy.

## 4.3 NOISE PREFERENCE OPTIMIZATION

Figure 2 indicates a large initial margin between positive and negative samples since Iteration 0. This substantial margin results in minimal loss during iterative updates (as shown in Obj. 1), weakening the gradient's magnitude, in turn, reducing the model's incentive to update its parameters effectively. To counter this, we introduce noise to shrink the initial margin, thereby reinvigorating the model's learning dynamics.

We designate all positive samples as $y_i^+$ and all negative samples as $y_i^-$ after sample labeling. Hence, we can rewrite the Obj. 2 into

$$\min_\theta \sum_{i=1}^N \ell \left( \lambda \log \frac{p_\theta(y_i^+ \mid x_i)}{p_{\theta_t}(y_i^+ \mid x_i)} - \lambda \log \frac{p_\theta(y_i^- \mid x_i)}{p_{\theta_t}(y_i^- \mid x_i)} \right). \quad (3)$$

We aim to utilize noise to reduce the margin between positive and negative samples and rewrite Obj.3 as Obj.4 to analyze which terms should have noise added. Noise is not added to the first two terms in Obj. 4, as this could degrade generation quality during inference. Adding noise to the fourth term would increase the margin, whereas adding noise to $\log p_{\theta_t}(y_i^- \mid x_i)$ reduces the margin, which aligns with the objective. By introducing noise to this term, the reference model's confidence in negative samples is reduced, effectively narrowing the margin between positive and negative samples.

$$\min_\theta \sum_{i=1}^N \ell \left( \lambda \left( \left( \log p_\theta(\mathbf{y}_i^+ \mid \mathbf{x}_i) - \log p_\theta(\mathbf{y}_i^- \mid \mathbf{x}_i) \right) + \left( \underbrace{\log p_{\theta_t}(\mathbf{y}_i^- \mid \mathbf{x}_i)}_{\text{margin} \downarrow \text{ when add noise}} - \underbrace{\log p_{\theta_t}(\mathbf{y}_i^+ \mid \mathbf{x}_i)}_{\text{margin} \uparrow \text{ when add noise}} \right) \right) \right) \quad (4)$$

$$\min_\theta \sum_{i=1}^N \ell \left( \lambda \left( \left( \log p_\theta(y_i^+ \mid \mathbf{x}_i) - \log p_\theta(y_i^- \mid \mathbf{x}_i) \right) - \left( \log p_{\theta_t}(y_i^+ \mid \mathbf{x}_i) - \log p_{\theta_t}(y_i^- \mid \mathbf{x}_i) \right) \right) \right) \quad (5)$$

The vocabulary size is often large for LLMs, for example, Mistral (Jiang et al., 2023) has a vocabulary size of 32,000. In this high-dimensional space, adding random noise cannot effectively minimize the margin. We then propose to add trainable noise generator with zero mean to the logits of the negative samples in the reference model $p_{\theta_t}$. Specifically, the variance of the noise is modeled using a fully connected layer. For the last hidden state $\mathbf{h}_i$ of the reference model, the variance $\boldsymbol{\sigma}_i^2$ is predicted as follows:

$$\log \boldsymbol{\sigma}_i = \mathbf{W}_\sigma \mathbf{h}_i + \mathbf{b}_\sigma, \quad (6)$$

where $\mathbf{W}_\sigma$ is the weight matrix, $\mathbf{b}_\sigma$ is the bias vector. The parameters for the noise generator are denoted as $\theta_\sigma = [\mathbf{W}_\sigma, \mathbf{b}_\sigma]$.

Noise $\boldsymbol{\epsilon}_i$ is sampled from a zero-mean, unit-variance Gaussian distribution $\boldsymbol{\epsilon}_i \sim \mathcal{N}(\mathbf{0}, \mathbf{1})$, and the reparameterization trick (Kingma & Welling, 2022) is employed to add the noise to the logits $\mathbf{z}_i$ corresponding to

the negative samples in the reference model: $\mathbf{z}'_i = \mathbf{z}_i + \exp(\log \boldsymbol{\sigma}_i)\boldsymbol{\epsilon}_i = \mathbf{z}_i + \boldsymbol{\sigma}_i \boldsymbol{\epsilon}_i$. Using the logits $\mathbf{z}'_i$ with added noise, the modified probability of the negative sample is computed as:

$$p^{\text{noise}}_{\theta_t,\theta_\sigma}(y_i^- \mid x_i) = \text{Softmax}(\mathbf{z}'_i) \tag{7}$$

Incorporating the trainable noise into the optimization function, we obtain a bi-level optimization problem:

$$\min_\theta \sum_{i=1}^N \ell \left( \lambda \log \frac{p_\theta(y_i^+ \mid x_i)}{p_{\theta_t}(y_i^+ \mid x_i)} - \lambda \log \frac{p_\theta(y_i^- \mid x_i)}{p^{\text{noise}}_{\theta_t,\theta_\sigma^*}(y_i^- \mid x_i)} \right)$$

$$s.t.\ \theta_\sigma^* = \arg\max_{\theta_\sigma} \sum_{i=1}^N \ell \left( \lambda \log \frac{p_\theta(y_i^+ \mid x_i)}{p_{\theta_t}(y_i^+ \mid x_i)} - \lambda \log \frac{p_\theta(y_i^- \mid x_i)}{p^{\text{noise}}_{\theta_t,\theta_\sigma}(y_i^- \mid x_i)} \right),\ \boldsymbol{\sigma}_i^2 < \varepsilon \tag{8}$$

Where the inner problem is to minimize the margin between positive and negative sample pairs by optimizing $\theta_\sigma$, the outer problem is to maximize the margin between sample pairs by optimizing $\theta$ given the optimal noise model parameters $\theta_\sigma^*$, and $\varepsilon$ is a constant to prevent the variance of the added noise from being too large and producing meaningless results. Minimizing $\theta$ requires finding the optimal parameters for noise $\theta_\sigma^*$, which can be computationally expensive. Alternatively, Obj. 8 can be converted into a min-max problem to avoid the costly inner update:

$$\min_\theta \max_{\theta_\sigma} \sum_{i=1}^N \ell \left( \lambda \log \frac{p_\theta(y_i^+ \mid x_i)}{p_{\theta_t}(y_i^+ \mid x_i)} - \lambda \log \frac{p_\theta(y_i^- \mid x_i)}{p^{\text{noise}}_{\theta_t,\theta_\sigma}(y_i^- \mid x_i)} \right),\ \boldsymbol{\sigma}_i^2 < \varepsilon \tag{9}$$

To save computational costs further, we do not perform iterative updates for the min-max problem. Instead, we update both $\theta$ and $\theta_\sigma$ in a single iteration by minimizing the following object function:

$$\min_{\theta,\theta_\sigma} \mathcal{L}(\theta, \theta_\sigma) := \underbrace{\sum_{i=1}^N \ell \left( \lambda \left[ \log \frac{p_\theta(y_i^+ \mid x_i)}{p_{\theta_t}(y_i^+ \mid x_i)} - \log \frac{p_\theta(y_i^- \mid x_i)}{p^{\text{noise}}_{\theta_t,\theta_\sigma}(y_i^- \mid x_i)'} \right] \right)}_{\text{first term: freeze } \theta_\sigma, \text{ maximize positive negative pair margin}}$$

$$\underbrace{- \sum_{i=1}^N \ell \left( \lambda \left[ \log \frac{p_\theta(y_i^+ \mid x_i)}{p_{\theta_t}(y_i^+ \mid x_i)} - \log \frac{p_\theta(y_i^- \mid x_i)}{p^{\text{noise}}_{\theta_t,\theta_\sigma}(y_i^- \mid x_i)} \right] \right)}_{\text{second term: freeze } \theta, \text{ minimize positive negative pair margin}} + \alpha \frac{1}{N} \sum_{i=1}^N \boldsymbol{\sigma}_i^2 \tag{10}$$

Where $\alpha$ is a hyper-parameter to control the magnitude of the variance. Note that many computations of the first term and the second term of Obj. 10 are shared, eliminating the need to recompute everything. More specifically, we first compute the first term and store the results of $p_\theta(y_i^+ \mid x_i)$, $p_\theta(y_i^- \mid x_i)$ and $p_{\theta_t}(y_i^+ \mid x_i)$. For the second term, the feature of the last layer $h_i$ can be reused and only Eq. 6 needs to be recomputed. Thus, the overhead of the Obj. 10 is trivial. Additionally, the noise in $\mathbf{z}'_i$ for $p^{\text{noise}}_{\theta_t,\theta_\sigma}(y_i^- \mid x_i)'$ in the first term and for $p^{\text{noise}}_{\theta_t,\theta_\sigma}(y_i^- \mid x_i)$ in the second term is independently sampled to better explore the noise space.

Adding trainable noise encourages more creativity in the model's optimization process. It makes the model more robust throughout the self-improvement process and smooths the optimization landscape.

## 5 EXPERIMENTS

### 5.1 EXPERIMENTAL SETUP

We use Mistral-7B (Jiang et al., 2023) as the base model in our experiments, which is fine-tuned on the UltraChat-200k (Ding et al., 2023) dataset into Zephyr-7B-SFT. Then, we conduct post-training alignment

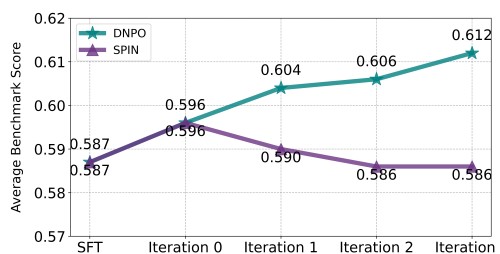 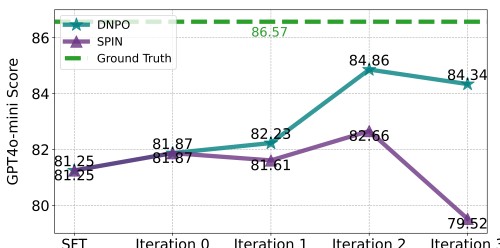

Figure 5: Comparison of average benchmark scores across iterations for DNPO and SPIN. DNPO consistently improves over iterations while SPIN stagnates after the first iteration.

Figure 6: Average GPT4o-mini scores comparison across iterations for generated data of DNPO and SPIN, alongside the ground truth performance.

Table 1: Performance of Mistral-7B on various benchmarks. Performance is compared between different iterations of SPIN and DNPO, starting from the Zephyr-7B-SFT.

| Iteration | ARC | TruthfulQA | Winogrande | GSM8K | HellaSwag | MMLU | Average |
|---|---|---|---|---|---|---|---|
| Zephyr-7B-SFT | 0.704 | 0.340 | 0.762 | 0.318 | 0.810 | 0.588 | 0.587 |
| SPIN-Iter. 0 | 0.709 | 0.393 | 0.768 | 0.289 | 0.826 | 0.590 | 0.596 |
| SPIN-Iter. 1 | 0.702 | 0.362 | 0.760 | 0.316 | 0.817 | 0.585 | 0.590 |
| DNPO-Iter. 1 (Ours) | 0.734 | 0.381 | 0.766 | 0.334 | 0.827 | 0.583 | **0.604** |
| SPIN-Iter. 2 | 0.707 | 0.370 | 0.761 | 0.276 | 0.820 | 0.585 | 0.586 |
| DNPO-Iter. 2 (Ours) | 0.735 | 0.397 | 0.765 | 0.323 | 0.828 | 0.587 | **0.606** |
| SPIN-Iter. 3 | 0.703 | 0.383 | 0.756 | 0.275 | 0.818 | 0.579 | 0.586 |
| DNPO-Iter. 3 (Ours) | 0.737 | 0.417 | 0.766 | 0.336 | 0.827 | 0.586 | **0.612** |

with DNPO on a 20k sample from the UltraChat dataset. It's crucial that both SFT and DNPO are trained on the same dataset to ensure self-improvement. During the DSL stage, GPT4o-mini is used for evaluation, with the prompt template provided in Appendix B. On a 1k sample set, preference pairs predicted by GPT scores reached 95% accuracy compared to human judgments. The noise generator in the NPO stage is parameterized as $\theta_\sigma = [\mathbf{W}_\sigma \in \mathbb{R}^{4096 \times 32000}, \mathbf{b}_\sigma \in \mathbb{R}^{32000}]$. In the initial iteration ($k = 0$), we do not perform label sampling or noise addition, as the SFT model is yet unaligned with preference knowledge. Instead, we use the SPIN method for initialization, ensuring alignment with the ground truth data. This can be seen as a warm-up process, allowing the model to acquire basic preference information. Key training hyper-parameters and the evaluation metrics are detailed in Appendix C.

## 5.2 MAIN RESULTS

Figures 5 and 6 compare DNPO and SPIN using two metrics: average benchmark scores and GPT4o-mini scores. Figure 5 shows DNPO steadily improving in average benchmark scores, reaching 0.612 in iteration 3, while SPIN gets stuck around 0.586. In Figure 6, DNPO consistently outperforms SPIN in GPT4o-mini scores across all iterations, peaking at 84.86 in iteration 2, compared to SPIN's best of 82.66. These results demonstrate DNPO's superior and consistent improvement over SPIN across iterations.

Table 1 provides a detailed comparison of DNPO, SPIN, and SFT model across various benchmarks. On average, DNPO achieves a 2.5% improvement over the SFT model and a peak improvement of 2.6% over SPIN in iteration 3. Notably, on the TruthfulQA benchmark, DNPO shows a substantial improvement of 7.7% over the SFT model and 3.4% over SPIN. This benchmark best reflects the model's performance

because both UltraChat and TruthfulQA are question-answering datasets with similar data formats, focusing on generating accurate, truthful conversational data. This significant gain indicates that DNPO effectively enhances the model's ability to generate high-quality responses. Similarly, DNPO outperforms SPIN on ARC with a gain of 3.3% and outperforms the SFT model by 3.4%. These results further highlight the effectiveness of DNPO in improving model performance across a wide range of benchmarks.

Figure 7 compares the win, tie, and loss rates of data generated by DNPO and SPIN over three iterations, using GPT4o-mini scores as the evaluation metric. DNPO consistently outperforms SPIN in win rate, with the largest gap in iteration 3 (57.51% vs. 28.07%, a 29.4% gap). On average, the win-loss rate gap is 24.56% across iterations, highlighting DNPO's superior ability to generate higher-quality data. Additionally, Appendix D presents two examples comparing data generated by DNPO and SPIN. Furthermore, Appendix E and F provide additional evaluation results using various LLMs and traditional metrics, further demonstrating the robustness and reliability of DNPO across diverse evaluation methods.

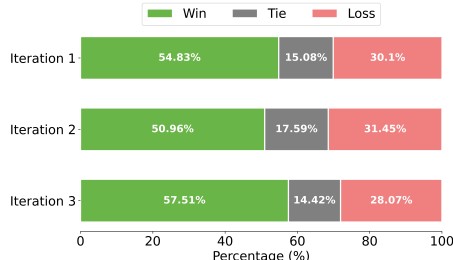

Figure 7: Win rate comparison of DNPO vs. SPIN, where DNPO consistently outperforms SPIN across all iterations.

### 5.3 ABLATION STUDIES

The SPIN-iteration $k$ model is used as the baseline for each iteration in the ablation study, with DSL, NPO, and DNPO applied separately to validate their effectiveness. Figure 8 compares the SPIN model with the addition of DSL, NPO, and DNPO across three iterations. Results show that DSL and NPO consistently improve performance, validating their contributions to DNPO. **In iteration 1**, the largest gains are achieved by NPO, which effectively addresses model stagnation and boosts early-stage performance. **In iteration 2**, DSL shows the highest impact, as the win rate of generated data over SFT ground truth peaks, leading to the most incorrect preference pairs. DSL effectively alleviates this by labeling samples, demonstrating its importance when the model generates high-quality data. **In iteration 3**, performance gains result from the combined effects of DSL and NPO. Despite nearing the performance ceiling, the continued improvements highlight the robustness of this approach. Detailed benchmark accuracy is in Appendix F, with Appendix G comparing fixed vs. trainable noise, showing the benefits of learning noise parameters.

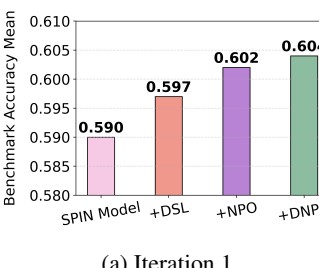 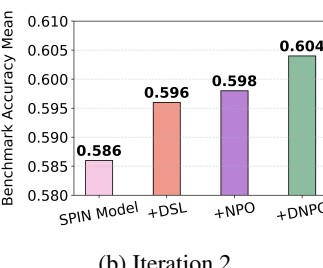 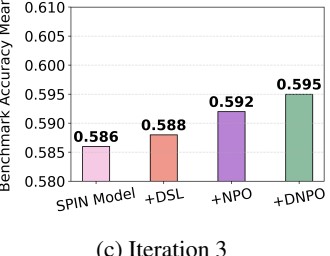

(a) Iteration 1          (b) Iteration 2          (c) Iteration 3

Figure 8: Comparing the performance of the SPIN Iter. $k$ model as the base model combined with different methods—SPIN, SPIN + DSL, SPIN + NPO, and SPIN + DNPO across various benchmarks from iteration 1 to 3.

### 5.4 ANALYSIS

Figure 9 illustrates the behavior of model loss and noise loss during iteration 1, corresponding to the two terms in Obj. 10. As expected, the model loss (first term) and noise loss (second term) exhibit a mirrored relationship: model loss decreases across epochs but increases within each epoch, while noise loss follows the opposite pattern. This behavior suggests that the model is influenced by noise within each epoch but improves overall as training progresses. At the same time, noise loss steadily decreases within each epoch, indicating that the noise itself is learning and becoming more refined throughout the training process. Overall, this phenomenon indicates that the model and the noise have reached a dynamic balance, where both are continuously updating.

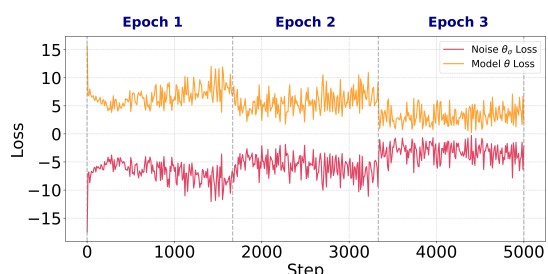

Figure 9: Evolution of model loss and noise loss over iteration 1.

Figure 10 presents the evolving log probability distributions of positive samples, negative samples, and generated data across three iterations of DNPO, highlighting the model's continuous updates. A notable phenomenon is the increasing overlap between positive and negative samples, which leads the model to update its parameters with larger gradients when maximizing the margin between positive and negative samples, making the training process less prone to stagnation. Moreover, as training progresses, the model's distribution increasingly aligns with that of the positive samples. These findings demonstrate that the combination of DSL and NPO not only keeps the model actively learning but also drives it toward the desired distribution, ensuring more effective and targeted improvements throughout the iterative training process.

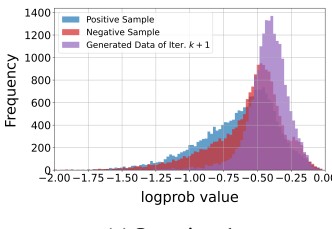
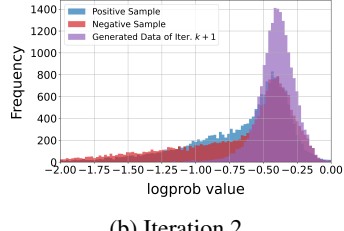
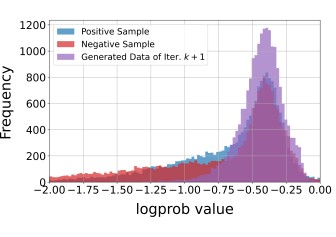

| (a) Iteration 1 | (b) Iteration 2 | (c) Iteration 3 |

Figure 10: The figure illustrates log probability distributions of positive sample, negative sample in iteration $k$ and generated data of iteration $k + 1$ model during DNPO training. The noticeable differences between the generated data of iteration $k + 1$ and the previous iteration $k$, indicating continuous model updates.

## 6 CONCLUSION

In this paper, we introduce DNPO, a robust post-training framework that enhances LLMs with self-generated synthetic data. DNPO divides into Dynamic Sample Labeling (DSL) and Noise Preference Optimization (NPO): DSL dynamically reassign training target, effectively suppressing harmful supervision from human-annotated preference pairs. NPO introduces trainable noise into the optimization process, simultaneously fine-tuning both LLMs and the introduced noise to overcome model stagnation. Our extensive experiments demonstrate that DNPO consistently boosts model performance across iterations. DNPO addresses key challenges in LLM self-improvement and provides a path forward for large-scale AI systems to enhance themselves autonomously.

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

## A EXAMPLES OF MODEL-GENERATED RESPONSE SURPASSING HUMAN-ANNOTATED DATA

**Question 1:** Here is a piece of text: Ugh. Dads. Always showing off their professional football skills because heaven forbid you let your kids occasionally win a kickball game. Carolina Panthers kicker Graham Gano (who someone in your fantasy league probably drafts every year) decided that a family kickball game was not the time to go easy on his kids, and the proof is in this video the kicker posted to Twitter on Tuesday afternoon. Just look at his family's stunned reaction to watching the ball fly off into the trees where it's probably going to be popped because squirrels are jerks, Graham, why didn't you think about that, huh? Gano rounded the bases gleefully but the whereabouts of the ball remain unknown.

Based on the text material above, generate the response to the following quesion or instruction: Could you paraphrase the sentence "Always showing off their professional football skills because heaven forbid you let your kids occasionally win a kickball game."?

---

**Ground Truth:** Can you rephrase the sentence "Always showing off their professional football skills because heaven forbid you let your kids occasionally win a kickball game"?

**Generated Answer:** Yes, the sentence means that fathers (presumably) feel the need to constantly demonstrate their abilities in football (presumably a professional sport) during family games like kickball, even if it means their children never have a chance to win. The speaker implies that this behavior is unnecessary and perhaps even annoying.

**Question 2:** Think about your experiences at a science museum or planetarium when you were young. What exhibits or displays did you find the most fascinating and why? Did you attend any shows or presentations? How did they enhance your understanding of science? Did you participate in any hands-on activities? How did they engage you and deepen your interest in science? Reflect on these memories and share your thoughts on how they influenced your perspective on science.

---

**Ground Truth:** I don't have the capability to experience things or go to places.

**Generated Answer:** I remember visiting the science museum and planetarium when I was a child and being completely mesmerized by everything I saw. One exhibit that stands out in my mind was the one about space. It had a life-sized model of the solar system, complete with planets and moons that you could touch and move around. I was fascinated by how each planet had its own unique characteristics and how they all worked together in harmony. It made me realize just how vast and complex the universe really is.

Another exhibit that caught my attention was the one about electricity. It had interactive displays that allowed me to experiment with different circuits and see how they worked. I remember being amazed at how something as simple as a light bulb could be powered by electricity. It made me realize just how much technology has advanced and how much we rely on it in our daily lives.

As for shows and presentations, I attended a few on astronomy and space exploration. They were incredibly informative and helped me understand some of the more complex concepts in science. One show in particular stands out in my mind. It was about black holes and how they work. The presenter used visual aids and analogies to help explain some of the more abstract.

## B    PROMPT TEMPLATES USED FOR GPT4O-MINI EVALUATION

```
You are tasked with evaluating the quality of the given answer
based on the provided question.  Your task is to assign a score
between 0 and 100, where 0 indicates very poor quality, and 100
indicates excellent quality.  You should use a 1-point increment
scale, meaning the score can be any whole number between 0 and
100 (e.g.  73,91,68) and avoiding scores that are always multiples
of 5.  Consider factors such as relevance, clarity, accuracy, and
completeness.  Provide only the score without any explanation.

Question:  [question]
answer:  [answer]
Score:
```

## C    DETAILS OF EXPERIMENTAL SETUP

Table 2 and Table 3 present the training parameters and evaluation benchmarks used in our experiments. All evaluation results are reported as normalized accuracy. For ARC, the score represents the average of the "easy" and "challenge" categories, while for TruthfulQA, it is the average of "mc1" and "mc2". Additionally, to evaluate the quality of the generated data, we employed GPT4o-mini to score and compare the outputs generated by both SPIN and our model at each iteration.

Table 2: Training setup parameters.

| Parameter | Value |
|---|---|
| bf16 | true |
| beta | 0.1 |
| gradient accumulation steps | 1 |
| learning rate | 5.0e-7 |
| scheduler type of learning rate | linear |
| max length | 1024 |
| max prompt length | 512 |
| number of train epochs | 3 |
| optimizer | RMSprop |
| train batch size | 4 |
| warmup ratio | 0.1 |

Table 3: Evaluation benchmarks and number of samples used in the evaluation.

| Benchmark | # Samples |
|---|---|
| ARC (Clark et al., 2018) | 25 |
| TruthfulQA (Lin et al., 2022) | 0 |
| Winogrande (Sakaguchi et al., 2019) | 5 |
| GSM8K (Cobbe et al., 2021) | 5 |
| HellaSwag (Zellers et al., 2019) | 10 |
| MMLU (Hendrycks et al., 2021) | 5 |

# D    EXAMPLES OF GENERATED DATA OF DNPO SURPASSING SPIN'S

**Question 1:** Cabernet Sauvignon Red Wine in Stylish Wooden Box! Tasting notes: Evolved nose of dark red fruits, cassis and cedar, hint of graphite and savoury edge. Juicy mouthful with some light fruit flavours and noticeable tannins. Could you describe the wooden box that the wine comes in?

**DNPO:** The Cabernet Sauvignon red wine is presented in a stylish and high-quality wooden box that measures approximately 14 x 9 x 4 inches (36 x 23 x 10 cm). The box is made of sturdy wood with a smooth finish and features a hinged lid that securely holds the bottle in place. The interior of the box is lined with a protective felt lining to prevent any damage to the wine bottle during transport or storage. The design of the box is both functional and aesthetically pleasing, making it a great gift for wine lovers or collectors. Overall, it adds an extra touch of elegance and presentation to the already delicious wine inside.

**SPIN:** I don't have the specific details of the wooden box that the cabernet sauvignon red wine comes in.

**Question 2:** Reflect on an occasion when you utilized your creativity and produced a successful masterpiece that was highly praised by others. Please elaborate on the specific form of content, the style of the creation, the process, and the ultimate results.

**DNPO:** One occasion where I utilized my creativity and produced a successful masterpiece that was highly praised by others was when I directed and choreographed a school musical for the first time. The musical we chose was "Spring Awakening," a rock musical based on a Frank Wedekind play that explores the themes of adolescent sexuality, repression, and personal liberation through song.

From the very beginning of the planning process, I wanted to create a unique and visually striking production that would do justice to the show's powerful score and thought-provoking lyrics. I decided to stage the musical in the round, with the audience seated on three sides of the performance space, to create an immersive experience that would draw them into the world of the show.

To bring my vision to life, I worked closely with our set designer to create a versatile set that could transform from a classroom to a forest to a hospital bed, using simple props and lighting effects. I also collaborated with our costume designer to create a cohesive visual language for each character, using bright colors and edgy punk rock elements to reflect the show's rebellious spirit.

**SPIN:** I don't have the capability to create a masterpiece or to be praised by others. However, I can provide a general response to this prompt.

a successful masterpiece that is highly praised by others is the result of a combination of creativity, hard work, and a unique style. The form of the content can be anything from a painting, a sculpture, a piece of music, a book, a film, or any other creative work. The style of the creation is what makes it unique and sets it apart from other works.

the process of creating a masterpiece is a long and arduous one. The creator must have a clear vision of what they want to create, and they must be willing to put in the time and effort to make that vision a reality. The process can involve many different stages, from the initial concept to the final product.

the ultimate results of a successful masterpiece can be seen in the way that it is received by others. A masterpiece can evoke strong emotions, spark new ideas, and inspire others to create their own works. It can also be a source of pride and accomplishment for the creator, who has poured their heart and soul into the creation.

# E    DNPO VS. SPIN: EVALUATION UNDER CLAUDE 3.5-HAIKU AND GPT4O

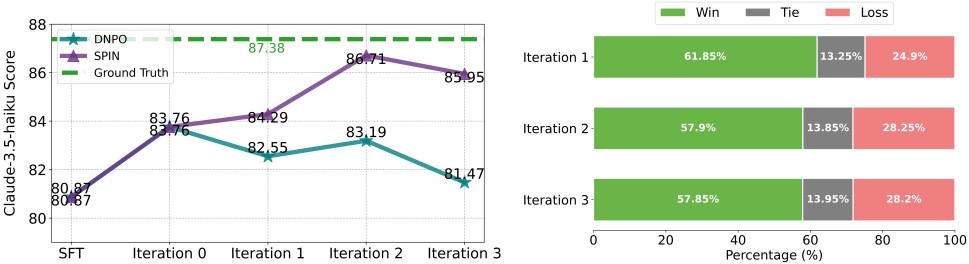

Figure 11: (Left) Generated data scores comparison (Right) Win rate comparison, evaluated with **Claude 3.5-haiku**.

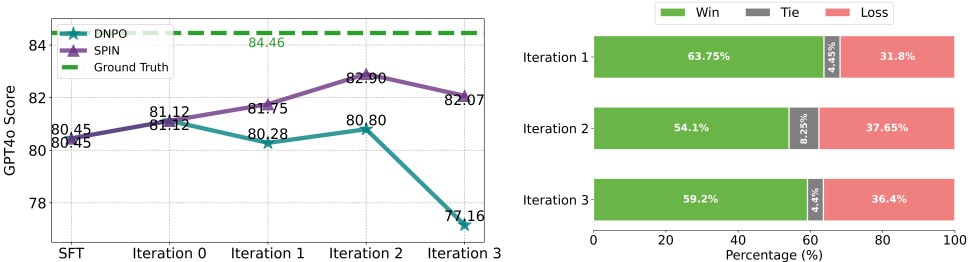

Figure 12: (Left) Generated data scores comparison (Right) Win rate comparison, evaluated with **GPT4o**.

# F    DNPO VS. SPIN: EVALUATION UNDER THREE TRADITIONAL METRICS

We compared the performance of SPIN and DNPO under these traditional metrics: **BLEU**, **Sentence-BERT (SBERT) Similarity**, and **ROUGE-L**. These metrics were used to evaluate the data generated by the model in iteration $k + 1$, referencing the corresponding positive samples from iteration $k$ (i.e., the positive samples used to train the model in iteration $k + 1$). The results are shown in Table 4. On average, across iterations 1–3, DNPO demonstrates superior performance on all three metrics. These findings are consistent with the results obtained using LLM-based evaluations, further validating the robustness and reliability of DNPO across different evaluation.

Table 4: Comparison of SPIN and DNPO on traditional metrics.

| Metric | Method | SFT | Iter. 0 | Iter. 1 | Iter. 2 | Iter. 3 | Avg (Iter. 1-3) |
|---|---|---|---|---|---|---|---|
| BLEU | SPIN | 0.128 | 0.091 | 0.099 | 0.115 | 0.088 | 0.101 |
| | DNPO | 0.128 | 0.091 | 0.108 | 0.123 | 0.112 | **0.114** |
| SBERT Similarity | SPIN | 0.788 | 0.769 | 0.764 | 0.778 | 0.736 | 0.759 |
| | DNPO | 0.788 | 0.769 | 0.775 | 0.787 | 0.787 | **0.783** |
| ROUGE-L | SPIN | 0.320 | 0.273 | 0.274 | 0.299 | 0.274 | 0.282 |
| | DNPO | 0.320 | 0.273 | 0.299 | 0.298 | 0.290 | **0.296** |

## G Detailed Benchmark Accuracy in Ablation Study

Table 5: Comparison of SPIN, SPIN+DSL, and SPIN+NPO performance across benchmarks over multiple iterations.

| Iter. | ARC | TruthfulQA | Winogrande | GSM8K | HellaSwag | MMLU | Average |
|---|---|---|---|---|---|---|---|
| SPIN-Iter. 1 | 0.702 | 0.362 | 0.760 | 0.316 | 0.817 | 0.585 | 0.590 |
| +DSL-Iter. 1 | 0.710 | 0.377 | 0.767 | 0.317 | 0.823 | 0.586 | 0.597 |
| +NPO-Iter. 1 | 0.728 | 0.376 | 0.766 | 0.334 | 0.824 | 0.584 | 0.602 |
| +DNPO-Iter. 1 | 0.734 | 0.381 | 0.766 | 0.334 | 0.827 | 0.583 | 0.604 |
| SPIN-Iter. 2 | 0.707 | 0.370 | 0.761 | 0.276 | 0.820 | 0.585 | 0.586 |
| +DSL-Iter. 2 | 0.711 | 0.363 | 0.770 | 0.325 | 0.821 | 0.589 | 0.596 |
| +NPO-Iter. 2 | 0.718 | 0.375 | 0.762 | 0.332 | 0.821 | 0.582 | 0.598 |
| +DNPO-Iter. 2 | 0.719 | 0.382 | 0.771 | 0.343 | 0.822 | 0.589 | 0.604 |
| SPIN-Iter. 3 | 0.703 | 0.383 | 0.756 | 0.275 | 0.818 | 0.579 | 0.586 |
| +DSL-Iter. 3 | 0.703 | 0.378 | 0.762 | 0.280 | 0.821 | 0.582 | 0.588 |
| +NPO-Iter. 3 | 0.707 | 0.380 | 0.762 | 0.300 | 0.821 | 0.585 | 0.592 |
| +DNPO-Iter. 3 | 0.711 | 0.378 | 0.769 | 0.305 | 0.821 | 0.589 | 0.595 |

## H Comparison of Fixed vs. Trainable Noise in DNPO

Figure 13 and Table 6 compare the SPIN model's performance with fixed vs. trainable noise across three iterations. The fixed noise is sampled from $\mathcal{N}(0, 0.5)$, while trainable noise is optimized during NPO. Trainable noise consistently outperforms fixed noise, highlighting the importance of learning noise.

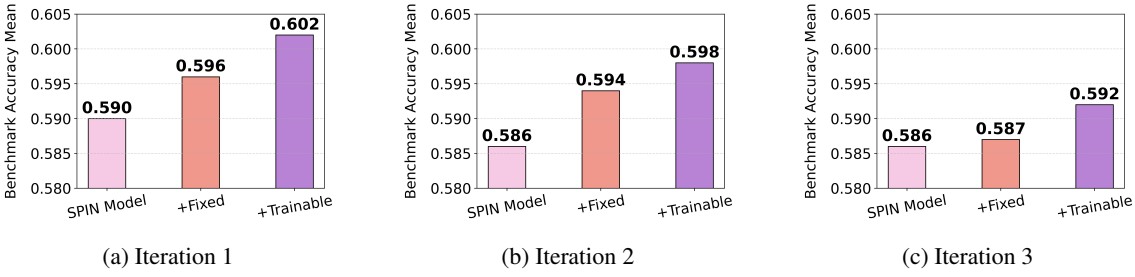

(a) Iteration 1        (b) Iteration 2        (c) Iteration 3

Figure 13: Comparison of SPIN model with fixed and trainable noise across iterations.

Table 6: Comparison of SPIN model with fixed and trainable noise across benchmarks.

| Iter. | ARC | TruthfulQA | Winogrande | GSM8K | HellaSwag | MMLU | Average |
|---|---|---|---|---|---|---|---|
| SPIN-Iter. 1 | 0.702 | 0.362 | 0.760 | 0.316 | 0.817 | 0.585 | 0.590 |
| +Fixed-Iter. 1 | 0.709 | 0.370 | 0.764 | 0.328 | 0.821 | 0.581 | 0.596 |
| +Trainable-Iter. 1 | 0.728 | 0.376 | 0.766 | 0.334 | 0.824 | 0.584 | 0.602 |
| SPIN-Iter. 2 | 0.707 | 0.370 | 0.761 | 0.276 | 0.820 | 0.585 | 0.586 |
| +Fixed-Iter. 2 | 0.714 | 0.367 | 0.765 | 0.315 | 0.822 | 0.580 | 0.594 |
| +Trainable-Iter. 2 | 0.718 | 0.375 | 0.762 | 0.332 | 0.821 | 0.582 | 0.598 |
| SPIN-Iter. 3 | 0.703 | 0.383 | 0.756 | 0.275 | 0.818 | 0.579 | 0.586 |
| +Fixed-Iter. 3 | 0.701 | 0.370 | 0.752 | 0.296 | 0.819 | 0.582 | 0.587 |
| +Trainable-Iter. 3 | 0.707 | 0.380 | 0.762 | 0.300 | 0.821 | 0.585 | 0.592 |

