# OpenReview forum: "Dynamic Noise Preference Optimization for LLM Self-Improvement via Synthetic Data"
_ICLR.cc/2025/Conference — Submitted to ICLR 2025_

### Official Review · Reviewer_jy4d · 2024-10-26

**Soundness:** 3
**Presentation:** 2
**Contribution:** 3
**Rating:** 6
**Confidence:** 2

**Summary:**

This paper attmpts to improve the LLM using generated data, and propose a method named DNPO. The authors challenge the assumption that the human-annotated data is superior than the generated data, and devise a dynamic smaple labeling strategy to pick both the human-annotated data and the synthesic data for LLM tuning. Ana an noise preference optimization is proposed to utilize the picked data from DSL. Experiment results show that the proposed method can attain better performance than SPIN.

**Strengths:**

using fake data to improve LLM is an interesting  and study-worhty problem.
a clear motivation is presented

**Weaknesses:**

1, Dynamic sampling labeling requires an "more powerful evaluation model"  to evaluate the human-generated data and synthesic data,   does this evaluation nedd to be superiror than the LLM model to be improved?  If yes, why do you not directly tune the evaluation model? If no, how do you ensure the given evaluation is credible and indeed pick more valuable data to fine-tune the LLM?

2, SPIN adopts 50K prompts from UltraChat for evaluation, while this paper only picks 20K,  why do not align the amount?

3,  How the amount of the synthesic data effects the final results, I did not observe any discussion regarding this quesion.

**Questions:**

Figure 2 x and y-axis are both not clear for me

---

> ### Author Response · Authors · 2024-11-23
> **Response to the Questions and Concerns raised by the Reviewer jy4d**
>
> Thank you for taking the time to review our work and provide feedback. We appreciate the detailed comments and the opportunity to address your concerns. Please find our responses to each of the issues raised below:
>
> **1.Role of the Evaluation Model in DSL:**
> We use a more powerful LLM as the evaluation model to ensure that the preference pairs are as accurate as possible. The evaluation model is specifically tasked with distinguishing between positive and negative samples, a critical step to minimize label noise and ensure that the generated data quality aligns with our training objectives.
>
> The evaluation model does not need to be inherently superior to the LLM being fine-tuned. Instead, its role is to provide a reliable and consistent measure of quality within the context of preference pair labeling. By leveraging a more capable evaluation model, we can increase the likelihood that the positive and negative labels accurately reflect true quality differences, which is essential for the success of our DSL method.
>
> Directly fine-tuning the evaluation model is not feasible in this context because the purpose of the evaluation model is to act as a static and consistent reference during the labeling process, rather than as a dynamic component that changes throughout training. Fine-tuning the evaluation model would risk introducing additional variability or bias, undermining its role as a reliable evaluator.
>
> **2&3. The amount of Training Data:**
> Our choice of 20k prompts was based on the constraints of our available computational resources. First, we view our algorithm as primarily designed for post-training optimization, where the data scale does not need to be as large as in pretraining. Second, our primary goal in this work is to demonstrate that DNPO outperforms SPIN and effectively addresses the two issues discussed in Section 3. Therefore, we focused more on these core contributions rather than the effects of data volume on the algorithm’s performance.
>
> Training on 20k data for one iteration—including data generation, DSL adjustment, training, and evaluation—takes approximately 10-12 hours on 4 A100 GPUs, which we consider a reasonable and efficient timeframe. Increasing the dataset size to 50k would raise the training time for one iteration to about 30 hours, and since both SPIN and DNPO experiments require 8 iterations in total, this would result in an impractically long runtime given our current resources and the rebuttal timeline.
>
> Unfortunately, we lack sufficient computational resources and time to explore the effects of dataset size during the rebuttal phase. However, we appreciate this insightful suggestion and will prioritize this investigation in future experiments to better understand the relationship between dataset size and model performance.

---

### Official Review · Reviewer_8RL5 · 2024-10-27

**Soundness:** 2
**Presentation:** 3
**Contribution:** 3
**Rating:** 6
**Confidence:** 2

**Summary:**

In this paper, the author intends to leverage the generated data to improve the performance of LLM.
Some assumptions are not well verified or have some conflicts.

**Strengths:**

The main idea is easy to follow.
The synthetic data is good to use.
The task on boosting LM without human annotations is good for the community to explore.

**Weaknesses:**

My main concerns are that some claims are not verified and ambiguous.

1. Some claims contain conflicts.
- "Is human-annotated data truly better? "
The author said that " around 30% of the generated data is of equal or higher quality compared to the human-annotated data."
More fairly,  according to the number in Figure 1, we also could say that around 80% of the real data is of equal or higher quality compared to the human-annotated data.
So human-annotated data is truly better than generated data.

- Only using GPT4o-mini as the data quality metric is questionable.
I think it would be better to use a fused metric with SSIM, FID and manual check.

- “stagnation” is ambigous.
Figure 2, I only can see the model overfits the generated data.
I think "overfitting" is better than using "stagnation".
How about more generated data, since the data generation is free from the human annotation?
Will the model still overfit the generated data?

2. Ablation studies.
The author claims that "the lack of variation in generated data across iterations, leading to stagnation during model updates."
One simple method is using the generative model for online data generation.

3. Contribution-1 Dynamic sample labeling (DSL)  is incremental.
DSL seems to be a simple filtering process?
I do not see any technical contribution. Similar filterring process has been widely used in the model trainning, such as BLIP.

4. The motivation of  Contribution-2 Noise preference optimization (NPO) is not well-proved.
Figure 10 is not supposed to be better than the existing methods in Figure 2.
I think the data distribution only shows the style similarity but does not indicate the good data for training.
Indeed, Figure 10 is simmilar to Figure 2, but the figure 10 shows more generated data.

**Questions:**

Please see the points in Weakness.

1. Some claims contain conflicts. Please explain.

2. Ablation studies are missing.

3. Contribution-1 is incremental.

4.The motivation of  Contribution-2 Noise preference optimization (NPO) is not well-proved.
Indeed, figure 10 is simmilar to Figure 2, but the figure 10 shows more generated data.

---

> ### Author Response · Authors · 2024-11-23
> **Response to the Questions and Concerns raised by the Reviewer 8RL5**
>
> Thank you for taking the time to review our work and provide feedback. We appreciate the detailed comments and the opportunity to address your concerns. Please find our responses to each of the issues raised below:
>
> **1. Conflict in Claims:**
>
> (a) We appreciate your observation regarding Figure 1. To clarify, our claim is not that synthetic data surpasses human annotations but rather that it reaches comparable quality in an nontrivial subset of cases (~30%). This indicates that the assumption of human-annotated data being inherently superior to generated data (made by concurrent works like SPIN) will introduce about 30% preference noise in every round.
>
> (b) Thanks for raising this concern. While SSIM and FID are image-related metrics for Computer Vision and not natively fit our setup, we believe that using multiple LLMs for data quality assessment is meaningful. Therefore, we have incorporated Claude 3.5-haiku and GPT-4o, two powerful LLMs, into our evaluation. The results are as follows:
>
> ### Performance Scores (Claude-3.5-haiku)
> The score of SFT ground truth data is **87.38**
> | Method       | SFT   | Iter0  | Iter1  | Iter2  | Iter3  |
> |--------------|--------|--------|--------|--------|--------|
> | **SPIN**     | 80.87 | 83.76  | 82.55  | 83.19  | 81.47  |
> | **DNPO**     | 80.87 | 83.76  | 84.29  | 86.71  | 85.95  |
>
> ### Win-Rate Comparison (Claude-3.5-haiku)
> | Iteration    | Win      | Tie      | Lose     |
> |--------------|----------|----------|----------|
> | **Iter1**    | 61.85%   | 13.25%   | 24.9%    |
> | **Iter2**    | 57.9%    | 13.85%   | 28.25%   |
> | **Iter3**    | 57.85%   | 13.95%   | 28.2%    |
>
> ### Performance Scores (GPT4o)
> The score of SFT ground truth data is **84.46**
> | Method       | SFT   | Iter0  | Iter1  | Iter2  | Iter3  |
> |--------------|--------|--------|--------|--------|--------|
> | **SPIN**     | 80.45 | 81.12  | 80.28  | 80.80  | 77.16  |
> | **DNPO**     | 80.45 | 81.12  | 81.75  | 82.90  | 82.07  |
>
> ### Win-Rate Comparison (GPT4o)
> | Iteration    | Win     | Tie      | Lose     |
> |--------------|----------|----------|----------|
> | **Iter1**    | 63.75%   | 4.45%    | 31.8%    |
> | **Iter2**    | 54.1%    | 8.25%    | 37.65%   |
> | **Iter3**    | 59.2%    | 4.4%     | 36.4%    |
>
> This further enhances the reliability of using LLMs for data quality assessment.
>
> (c) We apologize for the errors in the legends of Figures 2 and 10 in the previous version. The correct representation is as follows: red indicates negative samples, and blue indicates positive samples. Ideally, we aim for the distribution of the generated data to move closer to that of the positive samples (blue) while moving further away from the negative samples (red).
>
> However, in practice, we observe that the distribution of the generated data almost completely overlaps with the negative samples starting from the second iteration onward. This is why we describe the phenomenon as "stagnation" in model updates rather than "overfitting." Here, the model fails to improve or diverge meaningfully from the negative distribution.
>
> This observation motivated us to introduce trainable noise into our framework, addressing this stagnation by encouraging the model to break out of the overlapping distributions and continue improving its alignment with positive samples.
>
> **2.Ablation Studies:**
> Could you elaborate on how the generative model is used for online data generation? Does it involve generating more diverse data to enhance the dataset, or is it a process where the model generates data and trains on it simultaneously during updates?
>
> **3.Incremental Nature of DSL:**
> While DSL shares some similarities with traditional filtering techniques, its primary contribution lies in addressing the discovery that human-annotated data is not always superior to generated data. To mitigate the label noise introduced by this assumption, we developed DSL as a method to ensure the correctness of preference pair labels. By dynamically refining the labeling process during training, DSL maintains high-quality preference pairs, reducing the impact of noisy labels and enabling the model to learn more effectively from both generated and human-annotated data.
>
> Although DSL is a simple and effective method, it is not incremental. Instead, it challenges the traditional assumption that human annotation is always better. Furthermore, DSL is inseparable from the second component of our method, as together, they form the core of our innovation compared to other DPO-based approaches.

---

> ### Author Response · Authors · 2024-11-23
> **Following Response to the Questions and Concerns raised by the Reviewer 8RL5**
>
> **4. Motivation of NPO:**
> First, we clarify that the data volume in Figures 10 and 2 is the same, both comprising 20k samples. Second, in Figure 10, the positive samples are the truly high-quality samples identified through DSL, making them reliable labels. Our goal is for the generated data distribution to align more closely with that of the positive samples, while diverging from the negative samples.
>
> From the distributions shown in Figure 10, the generated data aligns significantly better with the positive samples compared to Figure 2. Furthermore, across the three iterations in Figure 10, the generated data distribution consistently moves closer to the positive sample distribution, indicating a trend of improvement. This contrasts with the stagnation observed in Figure 2.
>
> Therefore, the comparison demonstrates that NPO effectively improves the alignment of generated data with high-quality samples, validating its impact on enhancing data quality and training effectiveness.

---

> ### Comment · Reviewer_8RL5 · 2024-11-26
> **Thank you for the reply.**
>
> The author has addressed some of my concerns.
> Some problems remain.
>
> 1. First of all, the argument ` human annotation is always better` is not new.
> This is why we have label smooth and many different regularizations and noise mitigation.
> I do not think it is a contribution to support DSL.
>
> 2. Quality
> I still suggest to have some traditional metrics to validate the quality, which have been validated on many different tasks.
> Fairly, the proposed VLLM or MLLM evaluation is not widely-accepted in the community.
>
> 3. I do  not think DSL is specific for your argument. How about using other filtering method like BLIP?
> What is the difference of DSL from traditional filtering techniques?
>
> 4. Online data generation
> I guess the baseline fix the training data, which leads to overfitting.
> Could you generate different training data along with the training process?
> I mean the training data for the baseline model is not fixed. We always use ``unseen'' newly-generated data to train the model.
>
> 5. The figure can not support NPO.
> Sometime we get better distribution, which do not mean we suppose to have a better training effect.
> For instance, we need ``hard`` negatives not ``common`` negatives.
> Could you consider to use TSNE or other common data visualization tools instead of vanillia chart?
>
> Thanks a lot.

---

> > ### Author Response · Authors · 2024-11-27
> > **Response to the Remaining Concerns raised by the Reviewer 8RL5**
> >
> > Thank you for your response. Please find our responses to each of the issues raised below:
> >
> > **1.** In the context of post-training research where synthetic data is utilized for self-improvement, prior works consistently adopt a setup where SFT ground truth data is treated as the positive sample, and generated data is considered as the negative sample . This approach can be observed in works such as (https://arxiv.org/abs/2401.01335, https://arxiv.org/abs/2404.04291). However, the label noise referred to here arises from the mislabeling of positive and negative samples, which makes traditional methods like label smoothing ineffective in addressing this problem.
> >
> > Our approach using DSL is based on two key points:
> >
> > (a) When SFT ground truth is exclusively treated as the positive sample, the model in the post-training process continues to learn only the features of the SFT ground truth data. As shown in the results across all metrics, methods using this assumption (e.g., SPIN) suffer a significant performance drop in the final iteration. We attribute this to overfitting or collapse. By contrast, DSL dynamically adjusts the model’s labels during each iteration, creating a robust positive sample set composed of both SFT ground truth and generated data. Experimental results from DNPO and the ablation studies strongly support the effectiveness of this approach.
> >
> > (b) Our argument challenges the assumption that human-labeled data is always better when constructing preference pairs using LLM-generated data for self-improvement. Figure 1 experimentally demonstrates that for this task, human annotations are not always superior. Motivated by this observation, we propose DSL to reduce the risks of mislabeling preference pairs, thereby improving the robustness and reliability of model training. This perspective, coupled with the proposed solution, represents a novel contribution.
> >
> > **2.** We have taken your feedback into account and supplemented our evaluation with three traditional metrics: BLEU, Sentence-BERT (SBERT) Similarity, and ROUGE-L. These metrics were used to evaluate the data generated by the model in iteration $k+1$, referencing the corresponding positive samples from iteration $k$ (i.e., the positive samples used to train the model in iteration $k+1$).
> > We compared the performance of SPIN and DNPO under these traditional metrics, and the results are shown below:
> >
> > | Metric           | Method | SFT   | Iter. 0 | Iter. 1 | Iter. 2 | Iter. 3 | Avg (Iter. 1-3) |
> > |------------------|--------|-------|---------|---------|---------|---------|-----------------|
> > | **BLEU**         | SPIN   | 0.128 | 0.091   | 0.099   | 0.115   | 0.088   | 0.101           |
> > |                  | DNPO   | 0.128 | 0.091   | 0.108   | 0.123   | 0.112   | **0.114**       |
> > | **SBERT Similarity** | SPIN   | 0.788 | 0.769   | 0.764   | 0.778   | 0.736   | 0.759           |
> > |                  | DNPO   | 0.788 | 0.769   | 0.775   | 0.787   | 0.787   | **0.783**       |
> > | **ROUGE-L**      | SPIN   | 0.320 | 0.273   | 0.274   | 0.299   | 0.274   | 0.282           |
> > |                  | DNPO   | 0.320 | 0.273   | 0.299   | 0.298   | 0.290   | **0.296**       |
> >
> > On average, across iterations 1–3, DNPO demonstrates superior performance on all three metrics. These findings are consistent with the results obtained using LLM-based evaluations, further validating the robustness and reliability of DNPO across different evaluation methods.
> >
> > **3.** As addressed in the first question, DSL was designed specifically to tackle the issue of misleading noise that arises under the assumptions commonly used in the task of self-improvement with synthetic data. Unlike traditional filtering techniques such as BLIP, which is tailored for filtering captions in vision-language models by removing content unrelated to the primary subject of an image, DSL is fundamentally task-specific and addresses a different challenge: determining which sample, between SFT ground truth and generated data, should be the positive sample in a preference pair.
> >
> > In this context, we need an efficient method to construct labeled preference pairs without fine-tuning the evaluation model, as the focus is on rapid pairwise preference selection rather than optimizing the evaluation model itself. By leveraging a pretrained evaluation model to score two responses and select preferences directly, DSL provides a straightforward and effective solution.
> >
> > Moreover, DSL is tightly integrated with the DNPO framework we propose. Its design aligns with DNPO's objective of dynamically adjusting labels to mitigate noise and ensure robust training across iterations. This specificity and seamless integration make DSL distinct from traditional filtering techniques, as it not only resolves a task-specific issue but also enhances the overall performance and reliability of our framework.

---

> ### Author Response · Authors · 2024-11-27
> **Following Response to the Remaining Concerns raised by the Reviewer 8RL5**
>
> **4.** We appreciate your insightful suggestion and have indeed experimented with approaches to increase data diversity during training before. Specifically, we tried the following two methods:
>
> (a) **Training with Reward Model and PPO Algorithm:** We used a reward model (OpenAssistant/reward-model-deberta-v3-large-v2) and the PPO algorithm to train the model, which aligns with your idea of dynamically generating new training data during the training process. This method allows the model to update while simultaneously generating new data for training.
>
> (b) **Mixing Data from Iterations $k−1$ and $k−2$:** Inspired by this paper(https://arxiv.org/abs/2404.04291), we trained the model in iteration $k$ using a 50:50 mix of data generated by the models from iterations $k−1$ and $k−2$. This approach aimed to introduce diversity by leveraging data from different iterations.
>
> The results for iteration 1 are as follows:
>
> | Method   | ARC   | TruthfulQA | Winogrande | GSM8K  | Hellaswag | MMLU  | Average |
> |----------|-------|------------|------------|--------|-----------|-------|---------|
> | (a) | 0.714 | 0.352      | 0.754      | 0.271  | 0.788     | 0.567 | 0.574   |
> | (b) | 0.700 | 0.351      | 0.762      | 0.282  | 0.817     | 0.584 | 0.583   |
>
> Unfortunately, neither of these methods outperformed DNPO (the average score is 0.604). Additionally, we observed that using PPO for training presented significant challenges, including prolonged training times and instability due to the simultaneous generation of data and model training. As a result, we opted not to adopt these methods in favor of the more stable and efficient approach provided by DNPO.
>
> **5.** Thank you for your suggestion regarding the visualization methods. We emphasize that the improved training effects of DNPO are clearly demonstrated through the results in Table 1, Figures 5, 6, and 7, as well as the findings in the ablation study. These results collectively validate the superiority of DNPO in achieving better training outcomes.
>
> The comparison between Figure 2 and Figure 10 serves a different purpose. It provides an analysis of why DNPO outperforms SPIN. One potential reason we explored is the change in data distribution, which we visualized using the frequency distribution of $log$ probability of $p_\theta(y∣x)$ where $x$ is the question (prompt) , $y$ is the answer and $\theta$ is the current model. This metric was chosen as it reflects how data distribution aligns with the current model and serves as an indicator for evaluating distribution changes. The improved distribution in Figure 10, compared to Figure 2, illustrates a more favorable training trajectory under DNPO. This supports our hypothesis that the improved data distribution is a contributing factor to DNPO's success.
>
> If you have any further concerns, please let me know. Thank you!

---

> > ### Comment · Reviewer_8RL5 · 2024-11-29
> > **Thank you.**
> >
> > Thanks a lot for the further explanation.
> > It is good to see the comparison with PPO. It is quite important.
> >
> > 1. One key question still remains.
> > DSL is not specific for the task. I do not see the novelty or technical contribution for your contribtion on DSL.
> > The author reply does not solve my concerns. I think it is important but it is not a contribution.
> >
> > 2. Sorry. Why do not run the computer vision metrics, like FID, SSIM, Clipscore, as I suggested in the first reply?
> > They are widely used in GAN, Diffusion and other generative models.

---

> ### Author Response · Authors · 2024-11-30
> **Response to the Remaining Concerns raised by the Reviewer 8RL5**
>
> Thank you for your response. Please find our responses to each of the issues raised below:
>
> **1.** Our contribution lies in being the first to challenge the traditional assumption of "ground-truth labels are always better" in the context of self-improvement for LLMs. Through experiments, we demonstrated that this assumption introduces label noise in text generation tasks, leading to inaccuracies in preference pair annotations. To address this issue, we designed the DSL method, a novel and effective solution tailored to the characteristics of text generation tasks.
>
> **Text generation** exhibits a high degree of openness, particularly given that our training dataset, **Ultrachat**, is a **dialogue dataset** where generated outputs often allow for multiple correct expressions. This contrasts sharply with image tasks, where the correctness of targets is more explicit, and noise is typically the result of overt annotation errors rather than subtle quality differences between generated and ground-truth samples. Additionally, label-cleaning methods in image tasks often rely on fixed rules or features (e.g., static methods based on bounding boxes or pixel similarity), which are difficult to adapt to text generation due to its subjective correctness and reliance on context and semantics.
>
> The novelty of DSL lies in its ability to dynamically adjust positive and negative samples during training, fully accounting for the semantic complexity and contextual dependency of textual data. Unlike static methods that rely on predefined labels, DSL adopts an iterative and adaptive process, leveraging evaluation model prompting to dynamically clean labels and optimize preference pair annotations. This method is specifically designed for text generation tasks, and human verification has shown it achieves an annotation accuracy rate of up to **95%**.
>
> Moreover, compared to filtering methods that require additional training and hyperparameter tuning, DSL is more efficient and flexible in text tasks, requiring no extra training. Completing one round of preference pair labeling with DSL takes only **2-3 hours**, significantly reducing the overall training time of the framework. This allows computational resources to be focused on the core training phase of NPO. Ablation studies further validate the effectiveness of DSL, showing consistent positive contributions to model performance in every iteration.
>
> **In conclusion, the design of DSL fully considers the differences between text generation and image tasks, addressing the sources of label noise in text generation and the complexity of contextual semantics. By proposing a novel and efficient label-cleaning mechanism, DSL effectively resolves label noise in preference text pairs and significantly enhances framework performance. It offers a more flexible and streamlined solution for self-improvement in LLMs using text data.**
>
> **2.** Our training data and generated data are **purely textual**, so these CV metrics are not applicable. As mentioned in the previous response, we evaluated using three NLP metrics, which also reflect the evaluation philosophy of the CV metrics you suggested. For instance, Clipscore measures the semantic similarity between images and text, while SBERT calculates embeddings for predictions and ground truth to measure similarity. Therefore, we believe that LLM evaluations combined with traditional NLP metrics effectively demonstrate the superior quality of DNPO's generated data.

---

### Official Review · Reviewer_QJVE · 2024-11-04

**Soundness:** 2
**Presentation:** 2
**Contribution:** 2
**Rating:** 3
**Confidence:** 2

**Summary:**

This paper explores an approach that uses synthetic data to help train a LLM.  The authors argue that human annotated data is both expensive and noisy, and sometimes nosier than generated data.  As such, they use a LLM to help score generated vs human samples.  In addition, to help promote continuous improvement, the authors add additional noise to further boost performance.

**Strengths:**

1. The topic addressed is both important and timely
2. The solution proposed is intuitive, and makes sense
3. The specific implementation seemingly differs from some prior work on this same task

**Weaknesses:**

1. The authors only evaluate using a single architecture, and, thus, we don't know if the proposed improvements are only specific to this architecture.

2. The DNPO approach is quite similar to those in LNL, e.g., UNICON [A].  Whether the human or generated sample is the best label can also be framed from an LNL perspective.  As such, the contribution here can be argued as a simple application of this known solution.

3. Adding noise has some similarity to both masked language modeling and methods like excitation backprop [B] or self challenging [C].  While the proposed approach is seemingly different, it isn't clear if those differences are important

4. The gains are seemingly small enough that I would be concerned about their statistical significance.  The authors should provide a statistical test or even just error bars to alleviate this issue

5. The use of a LLM to evaluate the quality of the annotations is completely unconvincing.  One could argue that these language models are simply going to learn similar features, so all this test is doing is validating that this data is generated rather than that it is of a higher quality.

6. The authors cite several prior works that address the same task, but the authors do not compare against most of them.  As such, it isn't clear that the gains reported are significant in comparison to related work.

[A] UNICON: Combating Label Noise Through Uniform Selection and Contrastive Learning. CVPR 2022

[B] Top-down Neural Attention by Excitation Backprop. ECCV 2016

[C] Self-Challenging Improves Cross-Domain Generalization. ECCV 2020

**Questions:**

Unfortunately I cannot shortlist my weaknesses to fewer questions.  Each of them would have to be addressed for me to significantly raise my score.

---

> ### Author Response · Authors · 2024-11-23
> **Response to the Questions and Concerns raised by the Reviewer QJVE**
>
> Thank you for taking the time to review our work and provide feedback. We appreciate the detailed comments and the opportunity to address your concerns. Please find our responses to each of the issues raised below:
>
> **1.Single Architecture Evaluation:** While we focused on Zephyr-7B in our evaluations, we emphasize that most state-of-the-art LLMs, including Zephyr-7B, GPT-based models, and similar architectures, are decoder-only. This shared architecture minimizes the likelihood of our results being architecture-specific. Additionally, our choice of Zephyr-7B aligns with community practices for evaluating emerging methodologies without overextending computational resources.
>
> Similarly, SPIN also evaluates its methodology using only the Zephyr-7B model, and we chose to follow this practice to ensure consistency in comparison. However, we agree with the importance of evaluating across multiple models to generalize findings. To address this, we extended our experiments to Llama2-7B.
>
> Due to constraints in time and computational resources, we were unable to complete all iterations of these experiments. Completing one iteration, which involves data generation, preference sample evaluation, training, and assessment, requires approximately 18-20 hours with two A100 GPUs. As a result, we conducted comparisons for the first iteration of SPIN and DNPO on Llama2-7B. The win-lose rates for these experiments are as follows:
>
> |              | Percentage |
> |--------------------|------------|
> | **Win Rate**       | 45.2%      |
> | **Tie Rate**       | 19.35%     |
> | **Lose Rate**      | 35.45%     |
> | **Win-Lose Gap**   | 9.75%      |
>
> Note: The Win Rate reflects the percentage of cases where DNPO-generated data scored higher than SPIN in GPT4o-mini's evaluation.
>
> **2.Similarity to Prior Work (2&3):** We note that the referenced methods (UNICON [A], excitation backprop [B], and self-challenging [C]) originate from the computer vision domain and are fundamentally designed for image-related tasks. While there may be conceptual parallels, their applicability to LLMs is limited due to the distinct nature of textual data and sequence processing. DNPO, in contrast, is explicitly designed to address challenges unique to LLM training, such as iterative improvement and dynamic noise optimization for textual data, which are not the focus of the cited CV methods.
>
> Moreover, our method is built upon a detailed analysis of the DPO loss, a framework specifically designed for LLM fine-tuning, which the CV methods have not considered. In addition, our noise generator is tailored to the structure of LLMs, ensuring that the number of trainable parameters remains relatively small, thus preserving the efficiency and scalability of the approach. These design choices underscore the novelty and domain-specific focus of DNPO, setting it apart from methods designed for other modalities.
>
> **3.Comparison with Prior Work:** We acknowledge the importance of thorough comparisons with related work. Among the existing methodologies, SPIN stands out as one of the most notable approaches in this domain, making it a reasonable and meaningful benchmark for comparison. Moreover, other works differ from ours in their settings, as they do not use the data generated in the current iteration alongside the SFT ground truth. SPIN, being a relatively recent method with a setting closely aligned to ours, serves as an appropriate and relevant baseline for comparison. This alignment ensures that the evaluation is both fair and meaningful in demonstrating the advantages of our approach.

---

> ### Author Response · Authors · 2024-11-23
> **Following Response to the Questions and Concerns raised by the Reviewer QJVE**
>
> **4.Statistical Significance:** We understand the concern regarding the use of LLMs to evaluate annotation quality. To address this, we provide additional empirical evidence demonstrating the robustness and significance of our results. The standard deviations across various benchmarks on Zephyr-7B are as follows:
>
> | Dataset       | **ARC** | **TruthfulQA** | **WinoGrande** | **GSM8K** | **HellaSwag** | **MMLU** | **Average** |
> |---------------|---------|----------------|----------------|-----------|---------------|----------|-------------|
> | **Std**       | 1.1%    | 1.3%           | 1.2%           | 1.3%      | 0.3%          | 0.4%     | 0.9%        |
> | **Improvement compared to SFT**       | 3.3%    | 7.7%           | 0.4%           | 1.8%      | 1.7%          | -0.1%    | 2.5%        |
> | **Improvement compared to SPIN**      | 3.4%    | 3.4%           | 1.0%           | 6.1%      | 0.9%          | 0.7%     | 2.6%        |
>
> The average improvement in our approach, compared to the SFT model and SPIN model, is 2.5% and 2.6%, which far exceeds this range, indicating that our improvements are not random.
> Moreover, prior works, including SimPO (https://arxiv.org/pdf/2405.14734), report that their methods, along with various DPO variants, achieve average improvements of less than 1% compared to SFT on benchmarks similar to those we used. This highlights that achieving a meaningful margin of improvement across downstream tasks is inherently challenging. Our method’s consistent and significant gains across all evaluated tasks demonstrate that the improvements are substantial and not simply validating data generation but genuinely enhancing performance across diverse benchmarks.
>
> **5.LLM Evaluation Credibility:** We chose to use LLMs for evaluation primarily because our dataset comprises 20k samples, making human evaluation prohibitively time-consuming. Traditional metrics such as BLEU and ROUGE, while effective for certain tasks, fail to capture semantic relationships effectively in multi-turn dialogue datasets, which is why we opted for LLM-based evaluation.
> However, we acknowledge the potential concern of using the same LLM for both the DSL phase and the evaluation process, as this could lead to the model aligning with the specific preferences of that LLM rather than achieving broader improvements. To address this, we employed two other powerful LLMs, Claude 3.5-haiku and GPT-4o, to evaluate the generated data at each iteration. The results are as follows:
>
> ### Performance Scores (Claude-3.5-haiku)
> The score of SFT ground truth data is **87.38**
> | Method       | SFT   | Iter0  | Iter1  | Iter2  | Iter3  |
> |--------------|--------|--------|--------|--------|--------|
> | **SPIN**     | 80.87 | 83.76  | 82.55  | 83.19  | 81.47  |
> | **DNPO**     | 80.87 | 83.76  | 84.29  | 86.71  | 85.95  |
>
> ### Win-Rate Comparison (Claude-3.5-haiku)
> | Iteration    | Win      | Tie      | Lose     |
> |--------------|----------|----------|----------|
> | **Iter1**    | 61.85%   | 13.25%   | 24.9%    |
> | **Iter2**    | 57.9%    | 13.85%   | 28.25%   |
> | **Iter3**    | 57.85%   | 13.95%   | 28.2%    |
>
> ### Performance Scores (GPT4o)
> The score of SFT ground truth data is **84.46**
> | Method       | SFT   | Iter0  | Iter1  | Iter2  | Iter3  |
> |--------------|--------|--------|--------|--------|--------|
> | **SPIN**     | 80.45 | 81.12  | 80.28  | 80.80  | 77.16  |
> | **DNPO**     | 80.45 | 81.12  | 81.75  | 82.90  | 82.07  |
>
> ### Win-Rate Comparison (GPT4o)
> | Iteration    | Win     | Tie      | Lose     |
> |--------------|----------|----------|----------|
> | **Iter1**    | 63.75%   | 4.45%    | 31.8%    |
> | **Iter2**    | 54.1%    | 8.25%    | 37.65%   |
> | **Iter3**    | 59.2%    | 4.4%     | 36.4%    |
>
>
> The evaluation outcomes from these models align closely with those of GPT-4o-mini, providing strong evidence that the data generated by DNPO-trained models is of higher quality, independent of the biases of a single evaluation model. This reinforces the claim that DNPO improves data quality in a consistent and meaningful way.

---

> ### Comment · Reviewer_QJVE · 2024-11-28
>
> While I appreciate most of the experimental results, there are three concerns I wanted to flag:
>
> >LLM Evaluation Credibility: We chose to use LLMs for evaluation primarily because our dataset comprises 20k samples, making human evaluation prohibitively time-consuming.
>
> Couldn't you do a human study on a subset of samples if using all of them is too expensive?  Using other models certainly makes these experiments more robust, but it does not address the core limitation I pointed to.
>
> >We note that the referenced methods (UNICON [A], excitation backprop [B], and self-challenging [C]) originate from the computer vision domain and are fundamentally designed for image-related tasks.
>
> This is inherently untrue.  These methods may have been evaluated using image datasets, but they were selected by me since they introduce general concepts.  For example, self-challenging basically argues for robustness by adjusting the gradients using their strategy.  LLMs are also trained using gradient descent using loss functions, i.e., there isn't any technical limitation that would preclude their evaluation in this setting, and they have been used in other settings.  In fact, one could argue that if by evaluating on image datasets means that methods are restricted to use in those settings there would be little reason to have conferences like ICLR, as discussing ML topics across data types would not be possible.  Instead, each data type would have to silo their own conferences with little-to-no interactions between them.
>
> >Comparison with Prior Work....
>
> This was not a convincing argument.  For one, the authors could at least have a theorical argument why their approach was better, but in the related work section the authors do not even go that far.  In addition, if the authors feel that the methods are unfair, why not update prior work to make that comparison fair? And even if they are unfair, why not simply point to that as a factor as to why their approach performs better?  While making a fair comparison while also giving a general argument as to the specific contribution would be the best case, any of these would be better than ignoring it.

---

> > ### Author Response · Authors · 2024-11-29
> > **Response to the Remaining Concerns raised by the Reviewer QJVE**
> >
> > Thank you for your response. Please find our responses to each of the remaining issues raised below:
> >
> > **1.** As we discussed in our paper Section 5.1, before using LLMs as evaluation tools, we conducted a thorough validation by comparing LLM-predicted preferences with human-annotated preferences on a subset of 1,000 samples. This comparison showed a remarkable agreement rate of **95%**, which strongly supports the credibility of LLMs as reliable evaluation tools in our experiments.
> >
> > Additionally, to address the broader concern about relying solely on LLM-based evaluations, and considering feedback from reviewers about the potential limitations of this approach, we further supplemented our evaluation with three traditional metrics: **BLEU**, **Sentence-BERT (SBERT) Similarity**, and **ROUGE-L**. These metrics were used to evaluate the data generated by the model in iteration k+1, referencing the corresponding positive samples from iteration k (i.e., the positive samples used to train the model in iteration k+1). We compared the performance of SPIN and DNPO under these traditional metrics, and the results are shown below:
> >
> > | Metric           | Method | SFT   | Iter. 0 | Iter. 1 | Iter. 2 | Iter. 3 | Avg (Iter. 1-3) |
> > |------------------|--------|-------|---------|---------|---------|---------|-----------------|
> > | **BLEU**         | SPIN   | 0.128 | 0.091   | 0.099   | 0.115   | 0.088   | 0.101           |
> > |                  | DNPO   | 0.128 | 0.091   | 0.108   | 0.123   | 0.112   | **0.114**       |
> > | **SBERT Similarity** | SPIN   | 0.788 | 0.769   | 0.764   | 0.778   | 0.736   | 0.759           |
> > |                  | DNPO   | 0.788 | 0.769   | 0.775   | 0.787   | 0.787   | **0.783**       |
> > | **ROUGE-L**      | SPIN   | 0.320 | 0.273   | 0.274   | 0.299   | 0.274   | 0.282           |
> > |                  | DNPO   | 0.320 | 0.273   | 0.299   | 0.298   | 0.290   | **0.296**       |
> >
> >
> > The results of these evaluations consistently demonstrated the high quality of the data generated by DNPO. Moreover, DNPO outperformed SPIN across these traditional metrics, further reinforcing our confidence in the quality of DNPO's outputs.
> > By incorporating both traditional metrics and rigorous LLM validation, we believe our evaluation framework addresses the core limitations and provides a robust and multi-faceted assessment of model performance.

---

> > ### Author Response · Authors · 2024-11-29
> > **Following Response to the Remaining Concerns raised by the Reviewer QJVE**
> >
> > **2.** Thank you for raising these points. Let me clarify how our approach differs from the referenced methods and why we believe they are not directly applicable in our setting:
> >
> > **[A] UNICON: Combating Label Noise Through Uniform Selection and Contrastive Learning**
> >
> > One of UNICON's core motivations is to address class imbalance and label noise, which are not the primary focus of our work. DSL aims to generate high-quality preference pairs through a fast and straightforward approach, without relying on complex unsupervised feature learning processes (e.g., contrastive loss in UNICON). In our iterative framework, where preference pairs must be generated at every step, adopting UNICON would require repeated training in each iteration. This would significantly increase computational cost and make the preference pair construction process more cumbersome and time-consuming.
> >
> > In contrast, DSL uses an evaluation model to directly filter preference pairs without any additional training. This efficient and streamlined method is better suited to our requirement for quick preference pair construction in each iteration. While UNICON focuses on filtering clean data as its main goal, our approach treats this as just one step in a broader framework, emphasizing simplicity and efficiency in constructing preference pairs.
> >
> > **[B] Top-down Neural Attention by Excitation Backprop**
> >
> > I believe this paper has limited relevance to our work, as explained below:
> >
> > * **Conceptual Differences**: Excitation Backprop introduces a top-down attention model for CNNs based on a probabilistic Winner-Take-All process, focusing on identifying relevant regions in an image for classification. For instance, Figure 1 in that paper illustrates how this method highlights different areas of an image for various labels to improve classification. However, our work does not involve attention mechanisms or region-based analysis. Instead, our method focuses on introducing trainable noise to the loss function to adjust the margin between positive and negative samples, which is unrelated to the goals and techniques of Excitation Backprop.
> >
> > * **Architectural Differences**: Excitation Backprop is designed for CNNs, which differ significantly in structure and scale from the decoder-only architectures used in LLMs. The disparity in architectures and task objectives reduces the transferability and applicability of this method to our work. Additionally, the computational scale of LLMs makes certain techniques tailored for CNNs impractical, particularly when applied to tasks like language generation.

---

> ### Author Response · Authors · 2024-11-29
> **Following Response 2 to the Remaining Concerns raised by the Reviewer QJVE**
>
> **[C] Self-Challenging Improves Cross-Domain Generalization**
>
> We acknowledge conceptual similarities between Self-Challenging and our approach, as both introduce additional challenges to encourage robust learning. However, the methodologies and application domains differ substantially.
>
> * **Self-Challenging**: Drops features with high gradient norms at each epoch to construct harder feature sets for training, improving robustness in cross-domain generalization tasks (e.g., image classification).
>
> * **Our Approach (DNPO**): Introduces trainable noise to reduce the margin between positive and negative samples in the loss function, pushing the model to continue learning.
>
> This distinction highlights how we applied the principle of self-challenging to a different setting (CNN classification vs LLM self-improvement) and addressed domain-specific challenges, such as ensuring effective self-updating in generative models.
>
> We strongly believe in the value of cross-domain inspiration, as demonstrated by many seminal works:
>
> * **GANs**: Originally developed for image generation, GAN-style adversarial training has since been applied to natural language generation (e.g., Adversarial Generation of Natural Language(https://arxiv.org/abs/1705.10929), SeqGAN(https://arxiv.org/abs/1609.05473)).
>
> * **Few-Shot Learning**: Initially proposed for NLP tasks, few-shot approaches have since been adapted to computer vision and multimodal settings (e.g., CLIP(https://arxiv.org/pdf/2103.00020), Frozen Pretrained Transformers for Multimodal Few-shot Learning(https://arxiv.org/abs/2106.13884)).
>
> Moreover, we have experimented with selecting data based on the gradient value during propagation. Specifically, for our dataset of 20k samples, we selected the top 5k samples with the highest Fisher information (represented by the square of the gradient) and trained the model with them. However, as shown in our results at iteration 0 on Zephyr, this approach did not yield improvements compared to SPIN’s iteration 0 (the average score is 0.596) across benchmarks.
>
> | ARC  | TruthfulQA | Winogrande | GSM8K | HellaSwag | MMLU | Average |
> |------------|------------|------------|-------|-----------|------|---------|
> | 0.709      | 0.373      | 0.769      | 0.315 | 0.823     | 0.587| 0.596   |
>
>
>
> Finally, from the perspective of model architecture, applying Self-Challenging to our task would require masking features in the policy model during training. While this might have minimal impact on tasks like image classification, where full feature information is not always required, it could significantly affect language generation. The discrepancy between training and inference targets would likely lead to degraded performance, highlighting an inherent limitation in directly transferring this method.
>
> **3.** We supplemented our comparisons with related work (https://arxiv.org/abs/2404.04291) by evaluating an additional method that trains the model in iteration $k$ using a 50:50 mix of data generated by models from iterations $k−1$ and $k−2$, aiming to introduce diversity through leveraging data from different iterations (Method 1).
>
> Additionally, considering that both SPIN and this method use preference pairs and a loss function based on DPO, we explored an alternative training framework, PPO, which uses a reward model (OpenAssistant/reward-model-deberta-v3-large-v2) to assign explicit reward scores and continuously generate new data for training (Method 2). Both methods were tested on Zephyr-7B at iteration 1, and their performance on benchmarks was as follows:
>
> | Method   | ARC   | TruthfulQA | Winogrande | GSM8K  | Hellaswag | MMLU  | Average |
> |----------|-------|------------|------------|--------|-----------|-------|---------|
> | Method 1 | 0.714 | 0.352      | 0.754      | 0.271  | 0.788     | 0.567 | 0.574   |
> | Method 2 | 0.700 | 0.351      | 0.762      | 0.282  | 0.817     | 0.584 | 0.583   |
>
>
> Neither of these methods outperformed DNPO (the average score is 0.604). Additionally, we observed that using PPO required significantly more resources, taking approximately 2× the training time and 1.5× the memory of DNPO. These findings demonstrate DNPO's superior performance and efficiency, highlighting it as a more practical and effective method.
>
> If you have any further concerns, please let me know. Thank you!

---

> ### Comment · Reviewer_QJVE · 2024-12-02
>
> >One of UNICON's core motivations is to address class imbalance and label noise, which are not the primary focus of our work. DSL aims to generate high-quality preference pairs through a fast and straightforward approach, without relying on complex unsupervised feature learning processes (e.g., contrastive loss in UNICON). In our iterative framework, where preference pairs must be generated at every step, adopting UNICON would require repeated training in each iteration. This would significantly increase computational cost and make the preference pair construction process more cumbersome and time-consuming.
>
> >In contrast, DSL uses an evaluation model to directly filter preference pairs without any additional training. This efficient and streamlined method is better suited to our requirement for quick preference pair construction in each iteration. While UNICON focuses on filtering clean data as its main goal, our approach treats this as just one step in a broader framework, emphasizing simplicity and efficiency in constructing preference pairs.
>
> It seems that you are still focused too much on implementation rather than the idea.  You don't have to use the method exactly how it is proposed in a paper, but rather the idea behind it. In other words, I would argue that the core idea behind UNICON isn't the contrastive loss they use, but rather the re-labeling of samples.  As such, it isn't clear at all that using this component of the UNICON approach would require repeated training.  Your second paragraph seems to do a better job, but it seems to acknowledge the weaknesses I pointed to- that this specific component of your approach could be seen as an application of UNICON. It doesn't mean the whole framework is the same, but just the one component I specifically cited.
>
> >[B] Top-down Neural Attention by Excitation Backprop
>
> The majority of this response also conflates implementation differences with conceptual differences.  Like self-challenging, the idea behind excitation backprop and related papers like excitation dropout is simply to use gradient information to emphasize some features more than others.  Bringing up specific implementation concepts like "image classification" specific information does nothing to address the similarity in the ideas.  Your next response is better in this regard overall, but overall I would recommend the authors refocus their arguments on ideas as I have repeatedly been highlighting similarities in ideas and acknowledging differences in implementation myself.
>
> >Finally, from the perspective of model architecture, applying Self-Challenging to our task would require masking features in the policy model during training. While this might have minimal impact on tasks like image classification, where full feature information is not always required, it could significantly affect language generation. The discrepancy between training and inference targets would likely lead to degraded performance, highlighting an inherent limitation in directly transferring this method.
>
> I appreciate the majority of your response, but this comment seems to be, at best, conjecture, and would likely should have been skipped.  The number of language models that train using masked language modeling, which would follow the masking features idea, would bring into serious question whether this statement is true.  One could argue that it perhaps isn't the most effective training strategy, but to say it would lead to degraded performance outright is at best, a controversial position that would require extensive experimentation to be convincing especially in light of the widespread use of masked language modeling.  In other words, if I can train a language generation model by randomly masking some features and learning to predict the missing parts, why can't I do that if I focus on specific tokens in a non-random way? That's not to say that self-challenging would work for sure, it does sample things non-randomly and that could change the training dynamics in important ways (although a hyperparameter adjusting the non-random sampling would likely mitigate the worst effects), but the point the authors make here is clearly highly questionable and seems to contradict decades of language modeling research.

---

> > ### Author Response · Authors · 2024-12-03
> > **Response to the Reviewer QJVE**
> >
> > Thank you for your response.
> >
> > **1.** We acknowledge that, at a high-level conceptual perspective, DSL and UNICON do share some alignment in the re-labeling philosophy. Similarly, the ideas behind NPO and Self-Challenging also overlap in terms of deliberately increasing training difficulty to enable models to learn more knowledge. However, it is important to emphasize that these studies are predominantly conducted in the context of CNN frameworks, whereas similar exploration on LLMs is scarce. Considering the significant differences in architecture between LLMs and CNNs, particularly the much larger parameter scale of LLMs, transferring and validating these ideas in the LLM context represents a non-trivial exploration and novelty.
> >
> > Furthermore, we want to clarify that idea alignment does not equate to alignment in implementation. Different research efforts can implement the same idea in vastly different ways depending on the task, architecture, and requirements. In our paper, our goal is to demonstrate the effectiveness of DSL in the context of our task. Other implementations, such as UNICON’s contrastive loss, while valuable, are beyond the scope of our study. Additionally, our experimental results have shown that the implementation we propose works well in our task, further validating its suitability and practicality.
> >
> > **2.** Regarding the Self-Challenging method, we understand your key point that its core idea is to use gradient information to emphasize certain features, which can potentially be adapted to various tasks and frameworks. However, we would like to clarify the controversial statement we made earlier and elaborate on our viewpoint:
> >
> > For methods like DPO that involve training both a policy model and a reference model simultaneously, the reference model is absent during inference, and the policy model itself is dynamic during training. In such scenarios, introducing feature masking or noise into the policy model may lead to inconsistencies between training and inference targets, potentially causing instability. This is precisely why NPO chooses to add noise to the reference model rather than the policy model.
> >
> > It is important to clarify that we are not dismissing the effectiveness of masked language modeling. This method has been widely adopted for training language models and has achieved remarkable results. Our discussion focuses more on the stability issues that could arise when applying specific methods in certain tasks and frameworks. We fully agree that certain hyperparameters could mitigate potential instability to some extent, but this requires further experimental validation and is beyond the scope of this paper.
> >
> > **In conclusion**, we appreciate your detailed comments on the DSL approach and its relationship with other studies, such as UNICON and Self-Challenging. As you pointed out, there are many aspects worth exploring further, from the similarities in high-level ideas to the differences in specific implementations. We will continue to investigate how to better adapt these ideas to LLMs and contribute to advancements in the field.

---

### Meta-Review · Area_Chair_Qu4D · 2024-12-20

**Metareview:**

The paper proposes a method: Dynamic Noise Preference Optimization (DNPO) that uses synthetic data to improve an LLMs. This is done via the use of a dynamic sample labelling followed by a noise preference optimization. Results are then shown on a few different models in the main paper as well as the rebuttal. All the reviewers think that the topic the paper addresses is very important and they liked the use of synthetic data to improve the models. However, the reviewers did not find the rebuttal full satisfactory. They were not fully convinced by the contributions (originality vs instantiation) and some of the claims in the paper. Given the final rating, this paper unfortunately cannot be accepted to ICLR.

**Additional Comments On Reviewer Discussion:**

There was a robust discussion between the authors and 2 reviewers namely QJVE and 8RL5. However, reviewer jy4d did not participate in the discussion. After the multiple rounds of back and forth the reviewers not fully convinced that their concerns were addressed.

---

### Decision · Program_Chairs · 2025-01-22

Reject